# The Cultural and Commercial Value of Tulsi (*Ocimum tenuiflorum* L.): Multidisciplinary Approaches Focusing on Species Authentication

**DOI:** 10.3390/plants11223160

**Published:** 2022-11-18

**Authors:** Sukvinder Kaur Bhamra, Michael Heinrich, Mark R. D. Johnson, Caroline Howard, Adrian Slater

**Affiliations:** 1Medway School of Pharmacy, University of Kent Chatham, Chatham ME4 4TB, UK; 2Pharmacognosy and Phytotherapy, UCL London School of Pharmacy, Brunswick Square, London WC1N 1AX, UK; 3Chinese Medicine Research Centre, Department of Pharmaceutical Sciences and Chinese Medicine Resources, College of Chinese Medicine, China Medical University, Taichung 40402, Taiwan; 4Centre for Evidence in Ethnicity Health & Diversity, De Montfort University, The Gateway, Leicester LE1 9BH, UK; 5Tree of Life Programme, Wellcome Trust Sanger Institute, Wellcome Genome Campus, Cambridge CB10 1SA, UK; 6Biomolecular Technology Group, De Montfort University, The Gateway, Leicester LE1 9BH, UK

**Keywords:** *Ocimum*, *Tulsi*, *Holy basil*, medicinal plants, DNA barcoding, ITS, *trnH-psbA*

## Abstract

Tulsi (Holy basil, *Ocimum tenuiflorum* L., Lamiaceae), native to Asia, has become globalised as the cultural, cosmetic, and medicinal uses of the herb have been popularised. DNA barcoding, a molecular technique used to identify species based on short regions of DNA, can discriminate between different species and identify contaminants and adulterants. This study aimed to explore the values associated with Tulsi in the United Kingdom (UK) and authenticate samples using DNA barcoding. A mixed methods approach was used, incorporating social research (i.e., structured interviews) and DNA barcoding of *Ocimum* samples using the ITS and *trnH-psbA* barcode regions. Interviews revealed the cultural significance of Tulsi: including origins, knowledge exchange, religious connotations, and medicinal uses. With migration, sharing of plants and seeds has been seen as Tulsi plants are widely grown in South Asian (SA) households across the UK. Vouchered *Ocimum* specimens (*n* = 33) were obtained to create reference DNA barcodes which were not available in databases. A potential species substitution of *O. gratissimum* instead of *O. tenuiflorum* amongst SA participants was uncovered. Commercial samples (*n* = 47) were difficult to authenticate, potentially due to DNA degradation during manufacturing processes. This study highlights the cultural significance of Tulsi, despite a potential species substitution, the plant holds a prestigious place amongst SA families in the UK. DNA barcoding was a reliable way to authenticate *Ocimum* species.

## 1. Introduction

Tulsi, a herb renowned to have life prolonging and rejuvenating properties, is native to tropical Asia where it grows wild in warm regions [1]. It has been used in Asia, Africa and the Middle East for centuries, where it is widely incorporated in cuisine, cosmetics, herbal remedies and religious ceremonies [2]. Tulsi is part of various traditional medical systems including: Ayurveda, Siddha, and Unani. Ayurvedic scriptures refer to Tulsi as one of the main pillars of herbal medicine, first mentioned in the Rig Veda around 1500 BC [1]. Tulsi is one of the most sacred plants on the Indian subcontinent, as it represents a Hindu goddess, Virinda Tulsi. With the migration of people, the plant has become available around the world and grown in the UK. As the Tulsi plant travelled West, it became known to Christians as “holy” basil as is reflected in its Latin binomial, *O. *sanctum** L. (which is now recognized as a synonym of *O. *tenuiflorum** L.).

Tulsi is considered one of the most powerful herbs as it is used for treating a vast array of medical disorders, including conditions that affect the cardiovascular, endocrine, respiratory, and central nervous systems and the skin [3]. Traditionally, Tulsi has been used to treat anxiety/depression, asthma, bronchitis, diabetes, diarrhoea, eye disorders (chronic conjunctivitis, cataracts, and glaucoma), fever, insect bites, snake bites, malaria, and a variety of skin disorders [4,5]. Tulsi is also known to have antibacterial, antifungal, antiviral, antidiabetic, anticancer, antifertility, anti-inflammatory, adaptogenic, analgesic, cardioprotective, and hepatoprotective properties [6,7,8,9]. The use in chemotherapy and efficacy against the novel SARS-CoV-2 coronavirus, has also been explored [10,11]. The whole plant can be used, seeds, roots, leaves, flowers and even the stem [6]. Fresh or dried Tulsi leaves can be consumed, infused in teas, or used in cooking for systemic effects. Bathing in Tulsi infused water has been recommended for topical skin conditions such as eczema. A variety of commercial Tulsi products are also available including capsules, creams, juices, oils, shampoos, soaps, tablets, teas, and tonics.

DNA barcoding is a technique used to identify species based on the sequence of defined, short regions of DNA [12]. Variations in a defined region of the DNA sequence can discriminate between different species, and identify contaminants and adulterants [13,14]. The ideal DNA barcode region must be flanked by stable and highly conserved sequences for the use of universal primers in amplification and sequencing methods. The region between must be conserved within different individuals from the same species, but sufficiently variable to be able to distinguish between species [15]. DNA barcoding is a powerful and direct measure of species, enabling identification to the species level and beyond.

The genus *Ocimum* belongs to the family Lamiaceae. The genus has approximately thirty species including culinary basil (*O. basilicum* L.), and the Indian variety, Tulsi (*O. tenuiflorum*) [16]. There are a variety of different types of Tulsi available. According to Ayurvedic scriptures the most common are Raam and Shyam (also referred to as Krishna Tulsi). These two types are both classified as *O. tenuiflorum* whilst the less common Vana Tulsi is commonly assumed to be *O. gratissimum* (Figure 1), commonly known as African basil [17]. However, recent DNA and chemical analysis of the Vana Tulsi used in commercial teas (Pukka Herbs) shows it to belong to the haplotype characteristic of several basilicum-like *Ocimum* species [18]. Although the plants are characteristically different, they may all be referred to and kept as Tulsi [2]. Given the success of DNA barcoding for discrimination of *Ocimum* species, it could be used as an authentication tool to identify adulterants or species substitution [19].

The aim of this study was to investigate the cultural and commercial value of Tulsi (*Ocimum tenuiflorum*) amongst diasporic South Asian communities in the UK and to authenticate Tulsi samples using DNA barcoding techniques.

## 2. Results

### 2.1. Interviews with Participants Who Provided Tulsi Samples

In total, twelve interviews were conducted with South Asian participants from across three cities in the UK (Birmingham, Leicester and London). Two varieties of Tulsi were identified and kept by participants, including Raam and Shyam (Krishna) Tulsi (Figure 2).

Most participants (*n* = 11) grew their own Tulsi with varying degrees of success in the UK. Reference to how the cold climate with a dark winter period in the UK was not ideal for the growth and survival of Tulsi were noted. Tulsi seeds were often shared amongst family, friends, and religious institutes, both in the UK and abroad—from Africa and India. The sharing of seeds and cuttings was common practice amongst participants. Samples were also shared with the first author (SKB) for DNA authentication. A thematic analysis of the interviews identified several key themes (Figure 3) including types of Tulsi, knowledge transfer, cultural values and medicinal uses. The importance of Tulsi was attributed to the religious significance and numerous medicinal benefits, highlighting the cultural value of the plant. Participants mentioned the increased availability of commercial products available in Asian supermarkets and online which they once had to import from India but were now widely available in the UK.

### 2.2. Molecular Analysis—Creating Reference Barcodes

When this research project started there was a lack of reliable reference *Ocimum* sequences available in databases such as GenBank and the Barcode of Life Database (BOLD). Thus, a set of reference sequences were created to verify the species from a range of authenticated *Ocimum* samples that were sourced for this purpose. To create reference barcodes using the nuclear ribosomal internal transcribed spacer region (ITS) and *trnH-psbA* (Table 1), work was conducted at two institutes. Samples from sources P and C, Sample ID series G and V, became part of a collaborative study resulting in the submission of plastid sequences to GenBank. In that work, several plastid markers were sequenced and aligned, showing the species to cluster into four haplotypes [18]. Here, the ITS and t*rnH-psbA* regions of these same verified samples were sequenced, three reads were collated into contigs (Appendix A), and conflicts resolved to produce a consensus sequence. The *trnH-psbA* from *O. tenuiflorum* were further validated using the identification test published in the British Pharmacopeia [20]. This method requires that the *trnH-psbA* region is amplified and sequenced, and that this is aligned and compared with the reference sequence published. Once aligned, the sequences must match exactly five key regions of variation, carefully selected to prove the identification of the sample in question. All reference *O. tenuiflorum* samples passed this identification test.

The ITS sequences derived from these validated samples were aligned, along with others available in GenBank, and were found to cluster into three distinct groups (Appendix A). Interestingly, this nuclear marker followed the same clustering pattern that had been observed previously [18], with three of the plastid haplotypes (I–III) defined represented in this analysis. The alignment (Appendix A) shows the first 11 sequences aligning well together, representing the *gratissimum* group III, followed by 13 sequences corresponding to the tenuiflorum Group II and then Group I basilicum series. The final Group I is clearly more variable than the tenuiflorum and *gratissimum* groups. The ITS sequences of *O. africanum* (*G28*), *O. basilicum* (*G34*), *O. citridorum* (*G29*) and *O. kilimandscharicum* (G31) all feature in Group I, consistent with the clustering of these species within plastid haplotype I [18]. However, the variation within this group is high and the species level identification of samples within it is not guaranteed due to this.

The *gratissimum* Group III is interesting, because two sequences (G32 and C81) are quite distinct from the other *O. gratissimum* samples (including G30). This finding is consistent with haplotypes IIIa and IIIb defined in the plastid barcodes [18]. Within the *gratissimum* group, the two *O. selloi* sequences are also clustered (C82 and JF301405) although several sequence differences are observed.

#### 2.2.1. Authentication of Community Samples

Community samples were collected from interview participants and from participants who did not take part in interviews but volunteered samples of their Tulsi seeds/leaves; these formed the community Tulsi sample set (*n* = 26). The ITS and *trnH-psbA* regions from each of these samples was amplified and sequenced, and the resulting data aligned with sequences from the reference samples (Appendix A)

The species of 84% (*n* = 22) of the samples were identified using either one or both barcoding regions—ITS (*n* = 15) or *trnH-psbA* (*n* = 22) (Table 2) using the reference sequences created (Appendix A). Unsuccessful identification of some samples was due to: failed DNA extraction, poor sequencing data or contamination. The fail rate is linked to the region amplified, with the nuclear ITS region being less abundant than plastid markers and also longer, and therefore more prone to degradation. In total, 10 of the 22 samples identified were found to be *O. tenuiflorum*; of these, 5 were specified as Shyam by the donor, 1 Raam, and 4 were unknown. Eight of the samples were identified as *O. gratissimum*; of these, one was specified as Shyam by the donor, five Raam, and two unknown. These samples would be anticipated to align with the *O. tenuiflorum* samples, highlighting the issue of translating names for medicinal plants, which can lead to confusion and species substitution [21]. One sample (B03) was identified as a *Veronica* species, via GenBank, indicating contamination or substitution in the seed sample.

#### 2.2.2. Authentication of Commercial Samples

A range of commercial products (*n* = 47) were bought from around the UK, India, and online, including capsules, creams, soaps, syrups, shampoos, teas, tablets, oils, toothpaste, and a Tulsi rosary. Unprocessed Tulsi plant material used by Pukka Herbs^®^ was also included in this collection (*n* = 4). The ITS and *trnH-psbA* regions from each of these samples was amplified and sequenced, and the resulting data aligned with sequences from the reference samples. Just 5 commercial samples yielded DNA from which an ITS region could be amplified and sequenced, while the *trnH-psbA* region was amplified from 13 samples. Where these regions were amplified and sequenced, their identification could be assigned using the reference sequences (Table 3). The two samples sold as *O. basilicum* were identified within the basilicum Group, and all ten identified as Tulsi were identified as *O. tenuiflorum*. One final sample, described as ‘Lemon Vana’ was identified within the basilicum Group, and therefore not *O. tenuiflorum* or O. *gratissimum*. It was not possible to extract amplifiable DNA from the other commercial samples (*n* = 34). Particularly challenging were capsules, tablets, syrups, oils and cosmetics. This is likely due to the manufacturing and processing stages these samples have been through, which can eliminate or degrade DNA.

## 3. Discussion

### 3.1. The Cultural Significance of Tulsi

Tulsi still holds a prestigious position within UK based South Asian families and has major religious significance, especially for those of Hindu origin. Participants highlighted the plant is revered as a goddess and as such symbolises luck and good fortune if the plant is grown; hence, samples for authentication were widely available from participants. The medicinal uses were also important for participants in this study who used Tulsi for asthma, diabetes, eczema and other skin conditions [5,22,23]. It is evident the religious significance and medicinal properties of Tulsi have enhanced the cultural and commercial value of Tulsi [1,4].

### 3.2. Commercialisation of Tulsi: The Importance of Authentication

Participants mentioned the increasing ease of access to Tulsi products (as food supplements, medicines and cosmetics) available in the UK, which were previously unavailable or sourced from India. The increased availability of commercial products to meet consumer needs can increase the risk of adulteration due to the constant and rising demands of the market [24]. Plant species which are difficult to source, expensive, and have limited availability are subject to adulteration with products which maybe morphologically similar. Using taxonomically correct plants is critical to the safety and efficacy of herbal medicine [25]. Different species may have different pharmacological effects; therefore, the species used for commercial and medicinal purposes must be properly identified and authenticated. If Tulsi is kept for religious reasons, the botanical species of the plant may not be as important compared to if it was used for medicinal purposes. Fresh and dried Tulsi samples were often straightforward to authenticate, in comparison to commercial products bought, which were difficult to extract DNA from. This made it challenging to make conclusions of the efficacy of using barcoding for commercial products.

### 3.3. Evaluation of the Barcoding Regions Used to Authenticate Tulsi Samples

As recommended by The Consortium for the Barcode of Life (CBOL) Plant Working Group, initially the plastid *matK* and *rbcL* regions were explored for amplification of DNA samples [12,26]. The PCR results were highly variable, and the sequence data were poor. From the literature it was clear these regions were highly valuable; however, their use with the *Ocimum* species was limited and required optimization [18]. The nuclear ITS and plastid *trnH-psbA* were found to amplify and sequence consistently well in high quality samples [27]. Of the total 111 Tulsi and *Ocimum* samples collected during the research period, 82 samples were used for the final analysis, of which 67 samples (82%) were successfully identified with the combination of the ITS and *trnH-psbA* regions. Failure to identify some samples was due to unsuccessful DNA extraction or poor-quality sequencing data, this was more evident when working on the ITS region as it is less abundant and longer than the *trnH-psbA* region. Further work has since been conducted which has improved and validated the authentication processes [18,20,21,22,23,24,25,26,27,28].

The results of this investigation have enabled new reliable reference DNA sequences for the *Ocimum* species to be created and has contributed to DNA databases (Appendix A) [18]. This will enable more efficient authentication of *Ocimum* samples to take place. DNA barcoding is a useful technique for authenticating medicinal plants [18,20,29].

### 3.4. Identification of a Species Substitution

The two key types of Tulsi recognised in traditional Hindu texts are Raam and Shyam (Krishna) Tulsi which are considered to be varieties of *O. tenuiflorum* [17]. A number (n = 8) of Tulsi samples collected from participants in the UK, and one sample posted from South Africa, were identified using DNA sequence data as *O. gratissimum*. Of these 8 samples, 1 was specified as Shyam by the donor, 5 Raam, and 2 unknown, although all were provided as ‘Tulsi’, suggesting that a species substitution has occurred. Migration of SA communities from India to Africa, and then to the UK, may have affected the transfer of seeds and caused the use of an alternative species. *O. gratissimum* is native to Africa and known to grow abundantly even in the wild, whereas it is difficult to find in India and does not grow as well. Due to similar morphological features, it may have been misidentified by people or kept as a substitute as the preferred type was not available. Findings from the interviews for this research suggest Raam Tulsi was easier to germinate and grow as a houseplant in the UK in comparison to Shyam Tulsi. The results suggest species substitution has occurred for Raam Tulsi which appears to be *O. gratissimum* instead of *O. tenuiflorum*. The different species of Tulsi grown and used in the UK may be explained by the different migration and settlement patterns of SA communities and how well *O. gratissimum* grows in comparison with *O. tenuiflorum*.

According to the results from this investigation, Raam Tulsi species have been substituted in the UK, but how relevant this is to SA communities remains to be investigated. Many of the commercial samples collected did not specify the type of Tulsi (i.e., Raam or Shyam), except for samples from Pukka Herbs^®^; the Raam and Shyam Tulsi samples from Pukka Herbs^®^ did match the *O. tenuiflorum* references. In addition, Raam and Shyam samples from India were also correctly identified as *O. tenuiflorum;* further supporting the theory of species substitution in the UK.

## 4. Materials and Methods

A mixed-methods approach was used for this study; the use of semi-structured interviews and molecular techniques presented a dynamic approach to this multidisciplinary project. This study was part of a larger project looking at the use of herbal medicines by migrant SA communities in the UK [30].

### 4.1. Study Design and Participant Recruitment for Interviews

Semi-structured interviews, a type of qualitative research method which enables richer, in-depth opinions to be explored, were conducted with participants recruited from a previous study who disclosed they grew plants for medicinal purposes and agreed to take part in future studies [30,31]. A total of 19 eligible participants were identified and contacted, after which twelve interviews were conducted (October–December 2015) by the first author (SKB).

A participant information leaflet (PIL) was given to participants before written consent was obtained. The participant inclusion criteria included: anyone over the age of eighteen years old, of SA origin, who had a Tulsi plant or used Tulsi products. Responses were anonymised for data analysis. NVivo was used for thematic analysis.

The interview questions were designed to explore where participants had obtained their Tulsi from, if seeds were shared, grown and the type of Tulsi kept. Questions to explore the significance of keeping the plant such as cultural, religious and medicinal purposes were also included.

Tulsi leaves, seeds, and other samples were collected from participants during the face-to-face interviews for DNA authentication. Two interviews were conducted remotely via telephone; these participants posted samples to the researcher, in a pre-paid envelope. All Tulsi samples collected from participants were classified as ‘community samples’ and given a code to identify where they were collected from.

### 4.2. DNA Authentication of Tulsi Samples

#### 4.2.1. Sample Collection

A variety of samples were collected for molecular analysis over three years (2013–2015). This included Tulsi samples from participants taking part in the interviews and any volunteers encountered throughout the research period; these samples were labelled as ‘community samples’. Tulsi products (e.g., seeds, plants, tablets, capsules, oils, juices/syrups, soaps, and shampoos) bought from multiple sources in the UK, India and online were identified as ‘commercial samples’. The ‘reference specimens’ represent the reference *Ocimum* samples obtained from botanists and established sources (i.e., Professor Peter Nick (Botanical Institute Karlsruhe Institute of Technology, Germany), Dr. Eike Reich (Director at CAMAG Laboratory, Switzerland), and The Royal Botanic Gardens Kew—DNA bank (samples from Chase M.W and Suddee et al.). Samples were assigned a unique identification code followed by a chronological sample number: a letter indicating the source/location (A—Africa; B—Birmingham; C—commercial (C81–C85—purchased from Kew DNA bank); G—reference samples from Karlsruhe Institute of Technology, Germany; I—India; L—London; Li—Leicester; U—United states of America; V—reference samples from CAMAG.

#### 4.2.2. DNA Extraction

The Qiagen DNeasy Plant Mini Kit was used to conduct all DNA extractions. The Mini Protocol from the DNeasy^®^ Plant Handbook was used (Qiagen, 2012), starting with 0.1 g of fresh plant material or 0.02 g of dried material. NanoDropTM Lite (Thermo Fisher Scientific Inc) was used for quantification of DNA in the samples. A polymerase chain reaction (PCR) was conducted using a master mix (48 μL) comprised of 25 μL MyTaq Red Mix (2×), 21 μL distilled water, 1 μL forward primer and 1 μL reverse primer (Table 4) and 2 μL of template DNA. A positive control (known DNA sample) and negative control (distilled water) were always used to ensure the PCR had worked successfully and there was no contamination. Samples which gave a positive PCR result (as indicated by the gel electrophoresis) were sent for sequencing to an external sequencing company, Macrogen Europe, Amsterdam (The Netherlands) who provide Sanger sequencing services. Data were analysed using CLC Main Workbench 6 (CLC bio).

#### 4.2.3. Sequence Analysis

Good-quality DNA sequences traces were required, to produce a reliable contig, for analysis. For each sample, three reads (in the forward and reverse direction) were assembled and conflicts resolved by manual inspection of the traces (Appendix A).

DNA databases such as Genbank (http://www.ncbi.nlm.nih.gov/genbank/) and BOLD (http://www.boldsystems.org) were used to search for reference *Ocimum* sequences, which were compared with samples collected, when available. For some species more than one authenticated sample was available (i.e., from the reference samples and DNA sequences on GenBank) so these sequences were aligned to create references (Appendix A), used to authenticate unknown samples. Once reference sequences were created [18], they were aligned with trimmed sequences of the unidentified samples, in CLC, for visual inspection of similarities and differences (Appendix A). The Basic Local Alignment Search Tool (BLAST) (http://blast.ncbi.nlm.nih.gov) was also used to input the query DNA sequences, for identification of some samples.

## 5. Conclusions

The medicinal, cultural, and commercial value of Tulsi is rising as knowledge of its use is disseminated. Adulteration of plants used for medical purposes can have fatal consequences; therefore, the importance of selecting the correct species is of paramount importance. DNA analysis has proved to be a technique which can accurately identify and discriminate between the *Ocimum* species.

## Figures and Tables

**Figure 1 plants-11-03160-f001:**
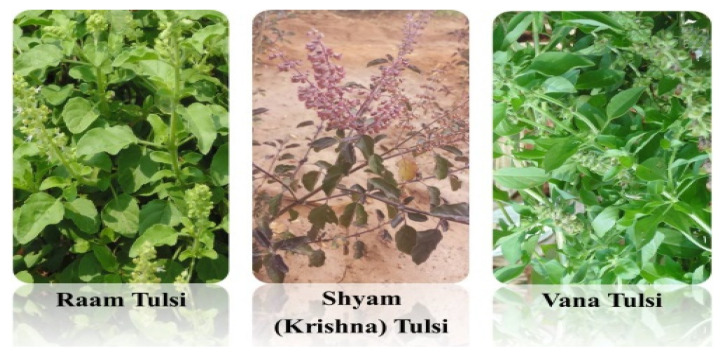
Raam, Shyam and Vana Tulsi grown in Bangalore, India (*pictures by S.K.Bhamra*).

**Figure 2 plants-11-03160-f002:**
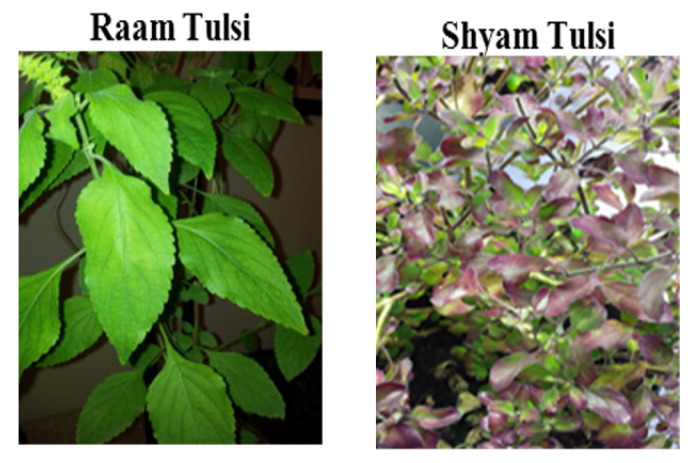
Raam and Shyam (Krishna) Tulsi grown in the UK (*pictures by S.K.Bhamra*).

**Figure 3 plants-11-03160-f003:**
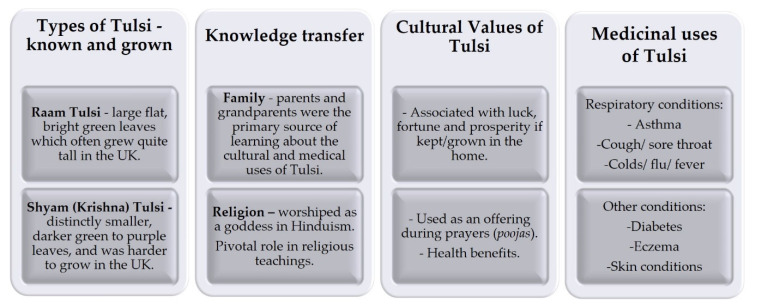
Key themes identified from thematic analysis of interviews with participants in the UK.

**Table 1 plants-11-03160-t001:** Reference Ocimum samples sequenced using ITS and *trnH-psbA* regions.

Sample ID:	Sample Source:	Type of Tulsi Specified:	ITS Sequence Obtained:	*trnH-psbA* Sequence Obtained:
G26	P	*O. tenuiflorum*	Y	Y
G27	P	*O. tenuiflorum*	Y	Y
G28	P	*O. africanum*	Partial	Y
G29	P	*O. citridorum*	Partial	Y
G30	P	*O. gratissimum*	Y	Y
G31	P	*O.kilimandscharicum*	Partial	Y
G32	P	*O. gratissimum*	Y	Y
G34	P	*O. basilicum*	Y	Y
G36	P	*O. tenuiflorum*	Y	Y
G37	P	*O. tenuiflorum*	Y	Y
G38	P	*O. tenuiflorum*	Y	Y
G39	P	*O. tenuiflorum*	N	Y
C81	K	*O. gratissimum*	Y	Y
C82	K	*O. selloi*	Y	N
C84	K	*O. tenuiflorum*	N	Y
C85	K	*O. americanum*	N	Y
V86	C	*O. tenuiflorum*	N	Y
V87	C	*O. tenuiflorum*	Y	Y
V88	C	*O. tenuiflorum*	Y	Y
V90	C	*O. tenuiflorum*	Y	Y
V92	C	*O. tenuiflorum*	N	Y
V93	C	*O. tenuiflorum*	Y	Y
V94	C	*O. tenuiflorum*	Y	Y
V96	C	*O. tenuiflorum*	N	Y
V98	C	*O. tenuiflorum*	Y	Y
V99	C	*O. tenuiflorum*	Y	Y

Key: P = Professor Peter Nick (Botanical Institute Karlsruhe Institute of Technology, Germany (G)); K = The Royal Botanic Gardens Kew—DNA bank (C); C = Dr. Eike Reich (CAMAG Laboratory, Switzerland (V)); N = No DNA sequence obtained; Partial = Low quality sequence data obtained, though sufficient to be cluster with the basilicum Group I (Appendix A); Y = DNA sequence successfully obtained.

**Table 2 plants-11-03160-t002:** Community Tulsi samples species identification summary.

Sample ID:	Tulsi Originally from (Donor):	Donor Location:	Reported Country of Origin of Tulsi:	Type of Tulsi Specified by Donor:	Species Identification—ITS:	Species Identification—*trnH-psbA:*
B01	Friend	Birmingham	-	Unknown	*O. gratissimum*	*O. gratissimum*
B02a	Friend	Birmingham	-	Raam	*O. gratissimum*	*O. gratissimum*
B02b	Aunty	Leicester	-	Shyam	*O. gratissimum*	*O. gratissimum*
B03	Mother	Amritsar	India	Unknown	*Veronica spp.*	*Veronica spp.*
B04	Friend	Birmingham	-	Raam	*O. basilicum type group*	*O. basilicum haplotype* I
B05	Friend	Leicester	-	Raam	*O. gratissimum*	*O. gratissimum*
B06	Friend	Leicester	-	Shyam	*O. tenuiflorum*	*O. tenuiflorum*
B06g	Friend	Leicester	-	Shyam	*O. tenuiflorum*	*O. tenuiflorum*
B07	-	-	-	Raam	*O. gratissimum*	*O. gratissimum*
B07g	-	-	-	Raam	*O. gratissimum*	*O. gratissimum*
L10	Cousin	India	India	Unknown	*O. tenuiflorum*	*O. tenuiflorum*
L11	-	-	-	Unknown	*O. basilicum type group*	*O. basilicum haplotype* I
A14	Mother-in-law	Durban, South Africa	Africa	Unknown	*O. gratissimum*	*O. gratissimum*
B15a	Temple	Birmingham	-	Raam	*O. gratissimum*	*O. gratissimum*
B15b	Temple	Birmingham	-	Shyam	*O. tenuiflorum*	*O. tenuiflorum*
Li79	Mother	Leicester	-	Unknown	*-*	*O. tenuiflorum*
Li80	Friend	Coventry	-	Basil	*-*	*O. basilicum haplotype* I
U100	-	-	America	Unknown	*-*	*O. tenuiflorum*
B102a	Mother	Gujarat	India	Shyam	*-*	*O. tenuiflorum*
B102b	Mother	Gujarat	India	Shyam	*-*	*O. tenuiflorum*
B102c	Mother	Gujarat	India	Raam	*-*	*O. tenuiflorum*
B102d	Mother	Gujarat	India	Unknown	*-*	*O. tenuiflorum*

Key: initial letter for sample ID indicates location of sample donor: A—Africa; B—Birmingham; L—London; Li—Leicester; U—United States of America. Letter after the reference number indicates multiple samples from the same donor.

**Table 3 plants-11-03160-t003:** Commercial Tulsi samples species identification summary.

Sample ID:	Sample Source:	Species/Type of Tulsi Specified:	Type of Sample Used for DNA Extraction:	Species Identification—ITS:	Species Identification—*trnH-psbA:*
B00	Basil plant from Tesco	*O. basilicum*	Fresh leaves	*O. basilicum*	*O. basilicum*
C08	Pukka three tulsi tea bag	Species not specified. *Raam, Shyam and lemon Tulsi*	Dried leaf material from a tea bag	*-*	*O. tenuiflorum*
C13	Tulsi seeds from ebay	*O. tenuiflorum* *(Shyam)*	Seeds—germinated. DNA extraction from leaves	*O. tenuiflorum*	*O. tenuiflorum*
C16	Akamba garden centre	*O. basilicum*	Fresh leaves	*O. basilicum*	*O. basilicum*
I44	Market in Fort Kochi, India	*-*	Dried leaves	*O. tenuiflorum*	*O. tenuiflorum*
I45	Phalada Agro farm	*O. tenuiflorum*	Dried leaves	*-*	*O. tenuiflorum*
I46	Phalada Agro farm	*O. tenuiflorum*	Dried leaves	*O. tenuiflorum*	*O. tenuiflorum*
I47	Phalada Agro farm	*O. tenuiflorum*	Dried leaves	*-*	*O. tenuiflorum*
I48	Phalada Agro farm	*O. tenuiflorum*	Dried leaves	*-*	*O. tenuiflorum*
C73	Pukka Herbs	Species not specified. *(Raam)*	Dried leaves	*-*	*O. tenuiflorum*
C74	Pukka Herbs	Species not specified. *(Raam)*	Dried leaves	*-*	*O. tenuiflorum*
C76	Pukka Herbs	Species not specified. *(Shyam)*	Dried leaves	*-*	*O. tenuiflorum*
C78	Pukka Herbs	Species not specified. *(lemon Vana)*	Dried leaves	*-*	*O. basilicum haplotype* I [18]

**Table 4 plants-11-03160-t004:** Primers and protocols used for amplification of barcode regions.

Region	Primer Sequence	PCR Protocol
ITS	ITS1 (Forward primer) 5′-TCCGTAGGTGAACCTGCGG-3′ ITS4 (Reverse primer) 5′-TCCTCCGCTTATTGATATGC-3′	Initial denaturation step—7 min at 95 °C. Followed by 30 cycles of 1 min at 95 °C, 30 s at 60 °C and 1 min at 72 °C. Final extension period—7 min at 72 °C.
*matK*	3F Kim (Forward primer) 5′-CGTACAGTACTTTTGTGTTTACGAG-3′ 1R Kim (Reverse primer) 5′-ACCCAGTCCATCTGGAAATCTTGGTTC-3′	Initial denaturation step—5 min at 94 °C. Followed by 5 cycles of 30 s at 94 °C, 40 s at 44 °C and 40 s at 72 °C. Then, 30 cycles of 30 s at 94 °C, 40 s at 46 °C and 40 s at 72 °C. Final extension period of 3 min at 72 °C.
*rbcL*	*rbcL*_f (Forward primer) 5′-ATGTCACCACAAACAGAGACTAAAGC-3′ *rbcL*_rev (Reverse primer) 5′-GTAAAATCAAGTCCACCRCG-3′	Initial denaturation step—5 min at 95 °C. Followed by 35 cycles of 30 s at 95 °C, 20 s at 52 °C and 50 s at 72 °C. Final extension period of 5 min at 72 °C.
*trnH-psbA*	*psbA* (Forward primer) 5′-GTTATGCATGAACGTAATGCTC-3′ *trnH* (Reverse primer) 5′-CGCGCATGGTGGATTCACAATCC-3′	Initial denaturation step—5 min at 95 °C. Followed by 35 cycles of 1 min at 95 °C, 30 s at touchdown temperature (touchdown temperature begins at 58 °C, reduced by 1 °C per cycle until 48 °C, then continued at 48 °C for the remainder of the program.) and 1 min at 72 °C. Final extension period of 7 min at 72 °C.

## Data Availability

Not applicable.

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
