# Peer review of "The Cultural and Commercial Value of Tulsi (Ocimum tenuiflorum L.): Multidisciplinary Approaches Focusing on Species Authentication"

_plants, 2022, doi:10.3390/plants11223160_

Round 1
Reviewer 1 Report
The study is well designed and article is well presented. I have two comments;
1.) In discussion part, the sub section titled “The prestige of Tulsi”, I would prefer the words like “the cultural significance of tulsi” rather than the word “Prestige”.
2.) Figure 3: it would be interesting to see if authors add another column as “cultural values of tulsi”.
Author Response
Dear Reviewer 1, thank you for your feedback and suggested changes. The amendments suggested have been made and we hope they meet your requirements.
Review 1
The study is well designed and article is well presented. I have two comments;
1.)In discussion part, the sub section titled “The prestige of Tulsi”, I would prefer the words like “the cultural significance of tulsi” rather than the word “Prestige”.
Authors response: This has been changed as per your suggestion.
2.) Figure 3: it would be interesting to see if authors add another column as “cultural values of tulsi”.
Authors response: This is a really useful suggestion. Figure 3 has been adapted to add a column on cultural values.

Reviewer 2 Report
This is a very interesting paper that brings together social sciences and molecular authentication to better understand the uses of Tulsi by South Asian families in the UK. The findings will be of interest to many in the field of ethnobotany and related fields. The story of the substitution of an African species for a South Asian one is especially interesting.
However, the barcoding methods are not adequately described.
The section 4.2.2 needs to be reviewed:
“The sequence data obtained were then used for analysis, identification, and construction of a reference database before authentication proceeded.” What does this mean? how can sequences be used for analysis and identification before the reference database is constructed?
“Data was analysed using CLC Main Workbench 6 (CLC bio). For each sample, three reads in the forward and reverse direction were assembled and conflicts resolved by manual inspection of the traces.” This seems to be back into sequencing methods, after previous sentence referred to “the sequence data obtained” but weren’t the same methods used for reference and test samples?
“The Basic Local Alignment Search Tool (BLAST) was used to input the query DNA sequences, for comparisons with reference DNA barcodes from reference databases for identification” I thought that the purpose here was to build a reference database, not to blast to existing ones? You say “DNA databases such as Genbank (http://www.ncbi.nlm.nih.gov/genbank/) and BOLD (http://www.boldsystems.org) were used to search for reference Ocimum sequences, which were then compared with query sequences in CLC.” Blast was used to compare to reference databases – plural – but didn’t you build your own reference database, so that it is unclear what or why you were blasting to these databases.
I suggest three separate subsections here
1. A section for the sequencing methods including assembly, assuming that the sequencing of references and test specimens used same methods.
2. A section about the reference database that must include reference to the specimens used, ideally the collector names and numbers in an herbarium collection, plus the genbank numbers because the sequences should be publicly available. The samples and genbank numbers are important. The authors should ensure that their new, reliable reference DNA sequences for the Ocimum species have been contributed to the DNA databases currently lacking this information. As they say, this will enable more efficient authentication of Ocimum samples to take place. It would be very useful to see some kind of analysis of these sequences so readers can see how similar or different sequences were between species, and how much variation there was within species.
3. Finally, a section about how the reference database was used to authenticate (identify) the test specimens, outlining what constituted an identification. This is particularly important since you state that just 82% of the samples were successfully identified. It is important to know what failure to identify means, is it when you have any sequence that is not a 100% match to a test sequence? Expanding on this in the discussion would be helpful, with reference to the analysis of sequences. Is there some other entity that people are cultivating in the UK that isn’t represented in your reference database?
I also have some minor suggestions that will improve the MS, as follows:
Abstract
“a range of coding regions (e.g. ITS and trnH-psbA)”
two regions a range? These are the regions that ultimately were used, so why refer to a range?
Introduction
“With the migration of people, the plant has become available around the world, and varieties are even grown in the UK.”
why varieties? - this has a very specific meaning in taxonomy - reword "available around the world and grown in the UK"
“(now referred to as O. tenuiflorum L.).”
which is now recognized as a synonym of O. tenuiflorum
“Part of the Lamiaceae family is the Ocimum genus, which has around thirty species”
I think this is a better way of writing: The genus Ocimum belongs to the family Lamiaceae. The genus has approximately 30 species…
Also, I found many more than 30 accepted names, so can you provide a reference source of the estimate of 30 names?
data is plural
Author Response
Dear Reviewer 2, thank you for your feedback and suggestions. The amendments suggested have been made and we hope they meet your requirements.
Comments and Suggestions for Authors
This is a very interesting paper that brings together social sciences and molecular authentication to better understand the uses of Tulsi by South Asian families in the UK. The findings will be of interest to many in the field of ethnobotany and related fields. The story of the substitution of an African species for a South Asian one is especially interesting.
Authors response: Thank you for the positive feedback on the overall paper it is highly appreciated.
However, the barcoding methods are not adequately described.
The section 4.2.2 needs to be reviewed:
“The sequence data obtained were then used for analysis, identification, and construction of a reference database before authentication proceeded.”
What does this mean? how can sequences be used for analysis and identification before the reference database is constructed?
Authors response: the section has been revised.
“Data was analysed using CLC Main Workbench 6 (CLC bio). For each sample, three reads in the forward and reverse direction were assembled and conflicts resolved by manual inspection of the traces.” This seems to be back into sequencing methods, after previous sentence referred to “the sequence data obtained” but weren’t the same methods used for reference and test samples?
Authors response: yes the same methods were used, this sentence has been moved up to improve the flow, as suggested.
“The Basic Local Alignment Search Tool (BLAST) was used to input the query DNA sequences, for comparisons with reference DNA barcodes from reference databases for identification” I thought that the purpose here was to build a reference database, not to blast to existing ones?
Authors response: both resources were exploited: our own reference sequences (validated by the work described by Jurges et al., 2018) and sequences that are now in GenBank and BOLD.
You say “DNA databases such as Genbank (http://www.ncbi.nlm.nih.gov/genbank/) and BOLD (http://www.boldsystems.org) were used to search for reference Ocimum sequences, which were then compared with query sequences in CLC.” Blast was used to compare to reference databases – plural – but didn’t you build your own reference database, so that it is unclear what or why you were blasting to these databases.
Authors response: Yes we did, the sequences for the plastid region have already been published (Jurges et al., 2018). I have tried to make this more clear using the subsections you have suggested.
I suggest three separate subsections here
- A section for the sequencing methods including assembly, assuming that the sequencing of references and test specimens used same methods.
Authors response: yes this was the same, this section has been made more clear and supplementary material (2 and 3) to reflect this has been added.
- A section about the reference database that must include reference to the specimens used, ideally the collector names and numbers in an herbarium collection, plus the genbank numbers because the sequences should be publicly available. The samples and genbank numbers are important.
- Authors response: this has been done in the table 2 which shows what reference samples were used from Genbank and the reference attached. A sentence has been added to reflect this in the methods too “Three Ocimum sequences from Genbank were used for comparisons ((DQ667240.1; JQ339256.1; FR726106.1).” Supplementary material 4, a multiple alignment of all reference sequences (personal collection and from GenBank) has been added to illustrate this.
The authors should ensure that their new, reliable reference DNA sequences for the Ocimum species have been contributed to the DNA databases currently lacking this information. As they say, this will enable more efficient authentication of Ocimum samples to take place.
Authors response: we agree this is very important. This is has been done for the plastid sequences and ITS are yet to be added.
It would be very useful to see some kind of analysis of these sequences so readers can see how similar or different sequences were between species, and how much variation there was within species. Authors response: supplementary material added (supplementary material added).
- Finally, a section about how the reference database was used to authenticate (identify) the test specimens, outlining what constituted an identification. This is particularly important since you state that just 82% of the samples were successfully identified. It is important to know what failure to identify means, is it when you have any sequence that is not a 100% match to a test sequence? Expanding on this in the discussion would be helpful, with reference to the analysis of sequences. Is there some other entity that people are cultivating in the UK that isn’t represented in your reference database?
Authors response: Section 3.3 amended to address this. Supplementary material 4 added to illustrate this too.
Sequences were identified both by BLAST searching of existing databases, alignment with reference sequences and (for O. tenuiflorum) matching with the reference sequence in the BP. BLAST searching of GenBank allowed us to identify the Veronica species that would not be identifiable if we had only used our reference database.
I also have some minor suggestions that will improve the MS, as follows:
Abstract
“a range of coding regions (e.g. ITS and trnH-psbA)”
two regions a range? These are the regions that ultimately were used, so why refer to a range?
Authors response: This has been amended in the abstract to now read as: “…and DNA barcoding of Ocimum samples using the ITS and trnH-psbA barcodes”
Introduction
“With the migration of people, the plant has become available around the world, and varieties are even grown in the UK.”
why varieties? - this has a very specific meaning in taxonomy - reword "available around the world and grown in the UK"
Authors response: This has been amended in the introduction to read as you have recommended.
“(now referred to as O. tenuiflorum L.).”
which is now recognized as a synonym of O. tenuiflorum
Authors response: This has been amended as suggested.
“Part of the Lamiaceae family is the Ocimum genus, which has around thirty species”
I think this is a better way of writing: The genus Ocimum belongs to the family Lamiaceae. The genus has approximately 30 species…
Authors response: This has been amended as suggested.
Also, I found many more than 30 accepted names, so can you provide a reference source of the estimate of 30 names?
Authors response: I am unsure of the source you are referring to. I have used the Medicinal Plants Names Service (Kew) and found over 400 non-scientific names. This has been added as reference/ hyperlink on page 5 Medicinal Plant Names Services | Kew
